# Cutaneous Reactions to Antidiabetic Agents: A Narrative Review

Aleia Boccardi and Jay H. Shubrook *

College of Osteopathic Medicine, Touro University California, 1310 Club Drive, Vallejo, CA 94592, USA;
aleia.boccardi@tu.edu
* Correspondence: jshubroo@touro.edu

**Abstract:** Diabetes is a common and complex disease affecting multiple organ systems throughout the body. With a consensus in care guidelines emphasizing the importance of glycemic control in determining the disease progression, people with diabetes worldwide have been placed on medication regimens targeting glucose stability from a variety of pathophysiologic pathways. Each of these medications also possesses its own potential for adverse events. In recent years, there has been increased reports of skin reactions to diabetes medications, adding to the more widely known eruptions such as insulin-induced lipohypertrophy and contact dermatitis of subcutaneous injections. The authors searched PubMed, Google, and Embase for articles including adverse reactions to anti-hyperglycemic medications. Key words and titles searched included, "antidiabetic drugs", "skin reactions", "adverse drug reactions", "allergic reactions", "diabetes", "metformin", "insulin", "DPP4 inhibitors", "thiazolindineones", "sulfonylureas", "SGLT2 inhibitors", "GLP-1 agonists", "diabetic medication", "injection site reactions". As a result, a total of 59 papers are included in this review. The great majority were case reports ranging from benign fixed drug eruptions to severe cutaneous reactions that threaten patients' lives. Increasing physician awareness of both the potential for, and presentation of, such reactions to diabetes medications can reduce hospitalizations and optimize care in an already vulnerable patient population.

**Keywords:** cutaneous adverse skin reactions; anti-diabetic medication; diabetes; drug reactions; rashes

## 1. Introduction

Approximately 463 million people in the world suffer from diabetes, with projections estimated to reach 700 million by 2045 [1]. Seated 9th in the World Health Organization's (WHO's) global leading causes of death, this noncommunicable disease can contribute to major destruction of the body's capacity to function [2]. With the advent of insulin 100 years ago, patients with diabetes found an avenue to fight the then universally fatal disease. Major studies of the following century, such as the United Kingdom Prospective Diabetes Study (UKPDS) [3] and Action in Diabetes and Vascular Disease (ADVANCE), Ref [4] have shown the importance of glycemic control in reducing microvascular sequalae of diabetes. Decreased saturation of glucose in the blood stream regulates the amount of intracellular glucose uptake, leading to less production of advanced glycation end products (AGEs) and their downstream effects. HbA1c less than 7% is generally recommended as a treatment target, but it is also important to maintain stable blood glucose levels. Regardless of moderate vs. tight glycemic control, data reveal improved outcomes over the long term with every 1% decrease in A1C [5]. Tackling the macrovascular sequalae requires a more multifactorial approach.

Given the efficacy of antidiabetic medication and the rising incidence of diabetes throughout the last two decades, pharmacological therapy has become a cornerstone for antidiabetic treatment regimens. Global pharmaceutical spending on this drug class alone rose by some USD 52 billion from 2008 to 2018 [6]. A patient-centered approach is vital in guiding the choice of antidiabetic medication based off efficacy, side effects, costs, comorbidities, and hypoglycemia risk. There are seven major classes of oral antidiabetic medications, each

with its individual side effect profile. However, aside from the normal gastrointestinal SE, a relatively recent emergence of cutaneous reactions is being witnessed throughout many of the major antidiabetic classes. This review provides a brief overview of cutaneous reactions to antidiabetic medications seen in case reports within the existing literature.

## 2. Brief Overview of Cutaneous Drug Eruptions

Toxidermia, otherwise known as cutaneous adverse drug reactions (CADRs), are skin manifestations of drug administration. While most are mild, there can be a broad range in visual characteristics of the eruptions, presenting as a singular erythematous plaque on the trunk to a full body sloughing of skin. Potentially life-threatening reactions, such as Stevens-Johnson syndrome (SJS), toxic epidermal necrolysis (TEN), drug rash with eosinophilia and systemic symptoms (DRESS), and acute generalized exanthematous pustulosis (AGEP) occur in 2% of CADRs [7]. The most common drugs implicated are antibiotics and anti-epileptics, with patients developing cutaneous complications in 1% to 5% of treatments [7]. In general, 2–3% of all adverse drug reactions arise in the skin [8].

The two basic classifications of cutaneous drug eruptions fall into immune and non-immune related reactions. Non-immunologic reactions account for the greatest portion, making up 75–80% of CADRs [7]. In the minority are the unpredictable effects, possibly immune-mediated reactions (20–25%), and the certain immune-mediated reactions (5–10%) [7]. Immunologic etiologies of CADRs are considered true allergies, resulting from a specific type of hypersensitivity reaction that can be immediate or delayed. Research into immunological pathways demonstrates previously uncovered mechanisms for allergic responses that may also play a role in CADRs, including T-cell stimulation [9]. Factors besides a primed immune system that can cause CADRs are linked to the drug's intrinsic properties, creating predictable reactions, drug tolerance, and pseudo-allergies.

Management of CADRs begins clinically with a detailed history and focused physical exam aimed at creating a comprehensive differential. Factors such as chronological timeline, morphologic features of the eruption, and systemic signs need to be taken into consideration. Withdrawal of the offending drug is the first line treatment for all CADRs. Methods to help identify the diagnosis include a biopsy for histological analysis of the lesions, or utilizing patch, prick, or intradermal testing for identification of a possible allergen. A critical step in management is to identify if the reaction pattern falls within the severe cutaneous adverse reaction (SCAR) patterns of drug hypersensitivity syndrome (DHS), Stevens-Johnson syndrome (SJS) or toxic epidermal necrolysis (TEN) through increasing familiarity with their wide-spread and rapidly progressive presentation, causative drugs, and clinical course. These diagnoses require aggressive therapy initiated as early as possible, with consideration of modalities to clear the offending agent from the system. Supportive measures can be added based on the severity of the eruption. For most mild cases, treatment with antihistamines, topical steroids, and emollients will aid in symptom management. Hospitalization for severe reactions is necessary to complete a thorough work-up and possibly administer intravenous immunoglobulin or systemic steroids.

Within the category of immune-related reactions, there are multiple subcategories including exanthematous, urticarial, vasculitic, blistering, pustular, and photoallergic [10]. The non-immune-related subcategories include nail changes, pigmentation changes, pseudoallergy, and selective cutaneous reactions [10]. Tables 1 and 2 demonstrate the important differences in each that help lead to accurate treatment and diagnosis.

**Table 1.** Immunologically Mediated Cutaneous Adverse Drug Reactions (CADRs).

| CADR | | Time to Presentation | Morphology | Distribution | Time to Resolution | Systemic Involvement |
|---|---|---|---|---|---|---|
| Exanthematous | Maculopapular eruption | 1 day to 3 weeks | Erythematous macules and papules, pruritic; range in size | Trunk, areas of trauma or pressure; spreads symmetrically to extremities | 1–2 weeks | No |
| | DRESS/DHS | 2–6 weeks | Pink to deep red maculopapular exanthem, can be accompanied by eosinophilia | Symmetrically arranged on face; spreads to body | 2–3 weeks | Fever, facial edema, adenopathies, visceral involvement |
| | Lichenoid | 2–3 months | Flat topped, erythematous-violaceous papules; eczematous/psoriasiform | Generalized over trunk and extremities | Weeks to months | Wickham striae usually absent |
| Urticarial | Urticaria/ angioedema | minutes to hours | Pruritic red wheals, hives, mucous membrane swelling | Variable | <24 h | Potential progression to anaphylaxis |
| Vasculitis | | 1–3 weeks | Palpable purpura ± petechiae, urticaria | Variable | Several days | Fever, arthralgias, facial swelling |
| Blistering | Bullous | 1–2 weeks initial episode 30 min–8 h subsequent | Well demarcated edematous plaques with erythematous-violaceous color; can progress to central dusky hue, bulla or erosion | Hands, legs, face, lips, genitalia, and oral mucosa | 1–2 weeks with prominent post-inflammatory hyper-pigmentation | No |
| | SJS: <10%. TEN: >30% | 7–21 days | Targetoid macules, mucocutaneous erythema with evolution to dusky plaques and full thickness sloughing; mucosal involvement | Trunk with spread to neck, face, and proximal upper extremities | Variable | Painful; fever, headache, and respiratory symptoms |
| Pustular | Acneiform | 1–3 weeks | Papular, pustular eruption without comedones | Face, trunk, extremities | A few weeks | No |
| | AGEP | <4 days | <5 mm non-follicular sterile pustules on erythematous skin | Face and intertriginous sites that generalizes in hours | 10 days | High fever, potential edema of face and hands |
| Photoallergic | | 24–72 h after exposure to light and agent | Red, scaling, pruritic papules and plaques | Photodistributed areas | Variable | No |

DRESS, drug reaction with eosinophilia and systemic symptoms; DHS, drug hypersensitivity syndrome; SJS, Stevens-Johnson syndrome; TEN, toxic epidermal necrolysis; AGEP, acute generalized exanthematous pustulosis. Adapted from Brockow et al. [11] and Alikhan et al. [12], and Schwinghammer et al. [13].

**Table 2.** Non-immunologic Cutaneous Adverse Drug Reactions (CADRs).

| ADR | Morphology | Distribution |
|---|---|---|
| Pigment changes | Hyper/hypo pigmented macules/patches; associated with melanonychia, oral pigmentation, lightening of hair | Localized or generalized; often photodistributed |
| Phototoxic | Sunburn-like eruption; potential blistering | Sun exposed areas |
| Irritant dermatitis | Acute: defined vesicular rash with necrosis; Chronic: dry, scaling, lichenification | Site of offending agent; hands most common |
| Pseudoallergy | Urticaria to angioedema, hypotension and anaphylaxis | Local or generalized |

Adapted from: Schwinghammer et al. [13].

## 3. Methods

The strategy utilized to collect relevant articles comprised of internet database searches within PubMed, Google, and Embase for articles involving adverse reactions to anti-hyperglycemic medications. Key words and titles searched included "antidiabetic drugs", "skin reactions", "adverse drug reactions", "allergic reactions", "diabetes", "metformin", "insulin", "DPP4 inhibitors", "thiazolindineones", "sulfonylureas", "SGLT2 inhibitors", "GLP-1 agonists", "diabetic medication", "injection site reactions". No date published parameters were set on the articles chosen due to the limited amount of data available on the subject. They ranged from 1986 to 2021, with all but 5 dated in the 2000s. Specifically, articles easily accessible in English, whether original or translated, were included, while all others were excluded. Resulting articles from the searches were screened for relevant information, with a total of 59 papers included in this review. Case studies were the most common type of publications found as noted in Table 3. Aside from self-database searches, two pharmaceutical companies, Sanofi and Lilly, were contacted regarding records for cutaneous reactions to their antidiabetic medications. A cascade method in article search was also utilized, searching through the "related articles" suggested on a page with an article of relevance.

**Table 3.** Breakdown of Articles.

| Studies | Total: 59 |
|---|---|
| Case reports | 41 |
| Case report + Mini-review | 5 |
| Lit. reviews: | 6 |
| Case Series | 4 |
| Research letter | 1 |
| Pooled analysis | 1 |
| Experiment | 1 |

## 4. Cutaneous Reactions to Antidiabetic Medication

### 4.1. Metformin

Metformin, a dimethylbiguanide and first-line agent for the treatment of type 2 diabetes exerts its mechanism of action through the inhibition of hepatic glucose output, decreasing intestinal absorption of glucose, and increasing insulin sensitivity. The FDA adverse reporting system (FAERS) details 3127 accounts of cutaneous adverse side effects, with pruritis, hyperhidrosis, and non-specific rash among the top three within this subcategory [14].

Most of the case reports for CADRs found in the literature related to metformin use fall within the category of immunologic reactions. Of this, six case reports fell within the

vasculitis subgroup as leukocytoclastic vasculitis (LCV), two in the blistering subgroup as fixed drug eruptions (FDEs), four in the exanthematous subgroup as rosacea-like rash, DRESS syndrome, psoriasiform and lichenoid drug eruption, and three photosensitivity reactions. No case reports were found to exhibit a non-immunologic pattern of CADRs. The most commonly reported reaction to metformin found was leukocytoclastic vasculitis (LCV). All six reports involved women from 33–60 years old and were biopsy proven LCV with a full work-up excluding other possible causes of LCV [15–20]. Described as hemorrhagic papules, vesicles and bullae, the vasculitis started on each of their legs and in some cases progressed to other areas of the body including trunks and forearms. Discontinuing metformin and starting prednisone significantly improved the eruption [15–20].

The cases of FDEs resulting from metformin varied from a generalized macular eruption with cutaneous hemophagocytosis to discrete erythematous lesions on the palms and soles of the patient [21,22]. Each resolved with metformin discontinuation, and like others in this family of drug reactions, recurred in the same location when exposed to the same medication. Reactions classified within the exanthematous immunologic CADRs were the broadest, covering benign rashes similar in morphology to rosacea and psoriasis to the life-threatening case of DRESS syndrome [23–25]. A rosacea-like facial rash occurred in a 29-year-old woman two days after starting metformin, thought to be the first non-vasculitis facial skin eruption due to this drug [23]. While a case of psoriasiform drug eruption secondary to metformin was reported in 2003, studies since have shown the drug used to both reduce psoriasis risk and improve its course [24].

The most serious cutaneous reaction to metformin involved a case of DRESS syndrome in a 40-year-old man [25]. Presenting with rash, pruritus, lymphadenopathy and eosinophilia after metformin, the patient quickly improved after drug withdrawal [25]. Photo-contact dermatitis was demonstrated in three patients, ranging from an eczematous to erythematous photodistributed rash that improved with the discontinuation of metformin [26].

### 4.2. Sulfonylureas

Sulfonylureas can be divided into first- and second-generation drugs that stimulate pancreatic beta cells to increase the release of insulin throughout the body. They are generally used in combination with metformin but can be considered for monotherapy in patients intolerant or unable to use metformin as a first line anti-hyperglycemic. The FAERS lists 3069 cases of skin and subcutaneous tissue disorders [14]. Toxic epidermal necrolysis and SJS are within the top 10 adverse events, reporting 125 and 115 cases, respectively, and are the most life-threatening [14]. Similar to metformin's CADRs profile, most of the sulfonylureas cause immunologic-related eruptions. However, different from metformin, the largest grouping within this category is a variety of loosely classified exanthematous reactions, including cases of TEN [27], psoriasiform rash [28], exanthematous pustulosis [29], pigmented purpuric dermatosis [30], and erythroderma [31], and lichenoid reactions [31–35]. Other types of immunologic reactions fell within the vasculitic group [36,37] and blistering reaction pattern (erythema multiforme) [38].

Only one case report of TEN was found in literature. A week after a 76-year-old patient started glimepiride, she developed a local pruritic skin rash that progressed to generalized erythema with multiple brown lesions and blisters and mucosal involvement [27]. Skin biopsy and a negative workup for other causes confirmed the diagnosis of TEN and the Naranjo probability scare for adverse drug reactions indicated that glimepiride-induced TEN was the probable cause. Another serious cutaneous reaction to sulfonylurea use, specifically gliclazide was found [31]. Erythroderma is a generalized exfoliative dermatitis involving greater than 90% of the body surface area that can present with pruritus, dyspigmentation, nail changes, and systemic findings. Most commonly caused from psoriasis, erythroderma can be induced from other etiologies including various drugs. In the latter cases, it presents with a shorter duration than other etiologies.

Pigmented purpuric dermatoses (PPD) represent a group of chronic diseases that clinically present as red to purple macules and patches with possible petechiae, and most commonly occur on the lower extremities. They are often asymptomatic but can be pruritic. The etiology of PPD is usually idiopathic, however, they can be caused by medications as seen in the case of glipizide-induced pigmented purpuric dermatosis [30]. Another benign subgroup of eruptions to sulfonylureas are the lichenoid reactions. These are uncommon rashes that can be difficult to distinguish from idiopathic lichen planus on histology. Clinically appearing as an eczematous or psoriasiform eruption in a photo distribution, it is more generalized than lichen planus and Wickham striae are usually absent [32–35].

Of note, phototoxicity to sulphonamide-derived oral anti-diabetes including glibenclamide, glipizide, glymidine, tolazamide, and tolbutamide was seen in a study using hairless mice [39]. After injection with test substances and radiation with UVA light, necrosis or edema was noted at readings 24 and 28 h later [39].

### 4.3. Meglitinides

Meglitinides are used in the treatment of diabetes and work by stimulating insulin release. The FAERS reporting system notes 577 cases of skin and subcutaneous tissue disorders in response to repaglinide and nateglinide [14]. Ninety-six cases of the 577 involve hyperhidrosis, with the next three most common reactions in this subgroup being pruritus, rash, and pemphigoid [14]. Only two instances of CADRs to meglitinides could be found in the literature, both of which can be categorized as immunologically mediated under exanthematous subtype. One case reports a 61-year-old male who developed a maculopapular rash after five days of treatment with repaglinide [40]. The article found was a post-surveillance study investigating the efficacy and safety of nateglinide use in combination with metformin. Adverse events involving the skin related to nateglinide use were rash and allergic dermatitis [41].

### 4.4. Glucagon-like Peptide-1 Receptor Agonists

GLP-1 agonists are a class of antidiabetic medication that work through activating GLP-1 receptors in the pancreas, stimulating insulin secretion and decreasing glucagon release. While they can be used as monotherapy if a patient has metformin intolerance, they usually are an adjuvant treatment if hemoglobin A1c continues to be uncontrolled. The FAERS database holds 9266 records of skin and subcutaneous tissue disorders arising after GLP-1 administration [14]. This is the largest number of adverse drug reactions relating to the skin after use of diabetes medication; however, only seven case reports have been found detailing instances of CADRs. All cases were immune related, with two exanthematous [42,43], and one of each urticarial [44] and blistering [45] reaction patterns noted in the literature.

The exanthematous reactions included a morbilliform drug eruption and generalized pruritic rash induced by dulaglutide and liraglutide, respectively [42,43]. Exenatide was seen to cause angio-edema in a 67-year-old woman who presented with tongue swelling, difficulty breathing, dizziness, and diffuse itching shortly after her injection [44]. This drug also was implicated in panniculitis in a 38-year-old woman that presented three weeks after starting exenatide [46]. In addition to these CADRs, two cases related to the drug's injection site were noted. The first involved a 35-year-old woman who presented with erythematous well-defined plaques surrounded by ecchymotic patches on the extensor surfaces of her thighs at the injection site [47]. The second case demonstrated exenatide-induced eosinophilic sclerosing lipogranuloma forming at the injection site of a 62-year-old patient [48]. Typically a consequence of high-viscosity fluid injected into tissues, it is seen most often during cosmetic procedures.

### 4.5. Sodium-Glucose Cotransporter-2 Inhibitors

SGLT-2 inhibitors are agents that lower glucose by blocking renal reabsorption of filtered glucose. These are all oral tablets taken once daily and are the newest oral medication for diabetes. The FAERS reports 4388 cases of skin and subcutaneous tissue disorders arising after SGLT2-I use, with diabetic foot, rash, and skin ulcers the top three adverse events accounting for 811, 765, and 617 of the total, respectively [14]. In 2016 a study was completed to determine the frequency and characteristics of hypersensitivity adverse events resulting from dapagliflozin. Overall, adverse events of hypersensitivity were low and not significant compared to placebo, with the most common events affected including rash, eczema, dermatitis, and urticaria [49]. Another study investigating the adverse effects of SGLT2-I assessed the post-marketing safety through systematically searching international pharmacovigilance databases. They found 1136 statistically significant skin and subcutaneous tissue disorders resulting from SGLT2-I use, with 7% of skin cases categorized as severe [50].

Several case reports existing in the literature discuss CADRs to SGLT2-I. The most commonly reported complication of SGLT2-I related to the skin is Fournier's gangrene. According to Bersoff-Match et al., 55 cases were reported by 31 January 2019, with two more found since [51–53]. The remaining cases fall within the immunologically-mediated subcategories of exanthematous (eczematous drug eruption) [54] and blistering (fixed drug eruption) [55] reaction patterns and one report of isolated generalized pruritis occurring in a 61-year-old woman [56]. Fournier's gangrene is a rare polymicrobial necrotizing fasciitis affecting the genital and anal areas. It presents with scrotal pain and redness which rapidly progress to fulminating gangrene. Diabetes mellitus predisposes patients to this disease on account of a weakened immune system. When associated with SGLT2-I, the infection is noted to occur anywhere between five days and 49 months, with prior urinary tract infections and morbid obesity considered predisposing factors [51].

### 4.6. Thiazolidinediones

Also known as the "glitazones," this medication class works to treat type 2 diabetes through acting as a nuclear transcription regulator and increases the sensitivity of tissues to insulin. No case reports in the literature could be found on skin eruptions associated with the thiazolidinediones drug class as of yet. However, data from the FAERS implicates the skin in 137 stated incidences [14]. Seventeen percent are attributed to episodes of urticaria, with hyperhidrosis accounting for the second largest subcategory of skin reactions, and erythema the third [14]. Other skin eruptions with multiple occurrences shown in the database include pruritis, alopecia, rash, blistering, angioedema, hyperkeratosis, palmarplantar erythrodysaesthesia syndrome, and dry skin. Below these listed, all other reactions have less than four reported cases [14].

### 4.7. DPP-4 Inhibitors

"Gliptin" drugs, are acting on the incretin pathway. Like GLP-1RA, they increase endogenous secretion, but they do so by blocking the enzyme that degrades GLP-1. These agents have had 248 cases of adverse skin and subcutaneous disorders reported in the FAERS system [14]. Pemphigoid reactions are seen to occur most frequently, with 76 of the cases attributed to this adverse event [14]. Cutaneous reactions discussed in the literature fall into the subcategories of immunologic CADRs exanthematous (generalized skin eruption [57], maculopapular eruption [58], and DRESS syndrome [59]), blistering (pemphigoids [60–67], SJS [68], TEN [68], fixed drug eruption [69]), hypersensitivity vasculitis [68], and photosensitivity [70] drug reaction patterns.

There is continuity between the FDA reporting system and what is found in the literature regarding DPP-4 inhibitors. Twenty-one presentations of bullous pemphigoid were found within case reports and case series, with the majority arising from the administration of vildagliptin [60–67]. Described as erythematous plaques and tense, pruritic blisters without mucosal involvement, drug-induced bullous pemphigoid presented anywhere between

2–37 months after medication initiation. Patients were treated by discontinuation of the medication and clobetasol. One case of non-bullous pemphigoid was demonstrated in a 70-year-old patient on linagliptin [67]. The patient presented with oral mucous membrane erosion and a few blisters on his upper chest and back that remitted once linagliptin was replaced with sitagliptin [67].

Other severe CADRs to DPP-4I were reported in a review of sitagliptin-associated drug allergies from 2006–2008 [69]. In their search, 48 cases were consistent with drug allergies with the most notable being two cases of SJS, two cases of TEN, and three presentations of vasculitis [68]. DRESS syndrome was also demonstrated to result from DPP4-I use, seen in a 67-year-old male with developed symptoms three months after starting vildagliptin [59].

### 4.8. Insulin

Out of the antidiabetic medications, cutaneous reactions to insulin have been the most studied. Throughout the years, formulations have been adapted to decrease the body's reactivity to certain ingredients, or types of specific insulins. The FAERS system reports a total of 8158 reactions to Insulin Aspart, Insulin Degludec, Insulin Detemir, Insulin Glargine, and Insulin Glulisine combined, with the top reactions pointing towards a type 1 hypersensitivity picture of pruritis, urticaria, and erythema [14]. Detailed extensively in the literature, insulin can incite multiple immunologic reactions in the skin, most falling in the urticarial reaction pattern. These include both local and systemic reactions of pruritus, urticaria, edema, and induration with progression to anaphylaxis [71]. Other types of skin-related reactions covered extensively are lipodystrophies that occur more in medium- and long-acting insulin preparations. Lipohypertrophy usually presents as soft, dermal nodules of varying size within the skin, and are thought to be from anabolic effects of insulin [72]. Lipoatrophy on the other hand is believed to be mediated through an immune response, presenting as an indentation from a loss of fat around the site of injection between 4 weeks and 2 years of use [73]. However, the latter reaction has decreased significantly since the replacement of bovine and porcine insulin with recombinant human insulin and analogue [73].

A lesser known skin-related complication of insulin therapy falls under the non-immunologic category of drug reactions: insulin-derived amyloidosis. Defined as a firm, singular subcutaneous nodule developing at the site of insulin injections, there have been more than 75 cases reported since its debut in 1983 [74]. Diabetics who are seen to be more susceptible generally have poor glycemic control at diagnosis with use of insulin from 4 to 60 years [74].

### 5. Conclusions

More than just local reactions at injection sites, there is an increasing number of case reports seen in the literature detailing various cutaneous eruptions identified to be a result of antidiabetic pharmacology. While rare, they range from type IV immunologic exanthematous reactions to non-immunologic pigment alterations that cease once the offending agent is stopped. The pathophysiology and classification of these reactions is not always known whether from patient refusal of biopsy or a vague histology pattern. The lack of information and familiarity with these drug side effects can pose a hurdle in diagnosis and treatment for both primary care providers and dermatologists alike. In addition, these drug reactions are not the first time the skin has been implicated in diabetes. The disease itself is associated with a range of cutaneous disorders, with 79.2% of diabetics experiencing some type of skin ailment [75]. The most common manifestations include cutaneous infections, xerosis, and inflammatory skin diseases [75]. Because they can appear at any time throughout the disease, from the initial presentation to the end, there exists room for potential hardships in determining the origin of the rash.

**Author Contributions:** Conceptualization, investigation, formal analysis, and writing—original draft preparation, A.B.; Methodology and visualization, A.B. and J.H.S.; Writing—review and editing, supervision, and project administration, J.H.S. All authors have read and agreed to the published version of the manuscript.

**Funding:** This research received no external funding.

**Institutional Review Board Statement:** Not Applicable.

**Informed Consent Statement:** Not Applicable.

**Data Availability Statement:** The data presented in this study are openly available in FDA Adverse Event Reporting System (FAERS) Public Dashboard.

**Conflicts of Interest:** Aleia Boccardi declares no conflict of interest. Jay H. Shubrook has served as a consultant to Abbott, Astra Zeneca, Bayer, Eli Lilly and NovoNordisk.

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
