# Peer review of "Cutaneous Reactions to Antidiabetic Agents: A Narrative Review"

_diabetology, doi:10.3390/diabetology3010008_

Round 1

Reviewer 1 Report

The manuscript entitled "Cutaneous Reactions to Anti-diabetic Agents: A Narrative Review". The author described the adverse cutaneous reaction as a result of various anti diabetic medications . The article would be a great deal of interest for researchers who are interested in designing studies related to dermal rashes induced by anti diabetic pharmacology. It is suggested to the author’s to make changes to limit the jargons where necessary. Best 

Author Response

Thank you for your review. Each of the authors have scanned through the manuscript to limit dermatologic and diabetes jargon. Thank you.

Reviewer 2 Report

The authors in the manuscript "Cutaneous Reactions to Anti-diabetic Agents: A Narrative Review" gave a brief overview of cutaneous reactions to anti-diabetic medications seen in case reports within the existing literature.

The review is informative, comprehensive, with extensive literature survey, clearly written.

Before acceptance, author should check the Abstract in the template. Namely, in the review from, the abstract represents the part of the introduction with literature. Please, correct in next submission.

Table 1, please add to the Table explanation of the abbreviation at the end of caption for the Table

Why did the authors through the text employ two abbreviations for the "Cutaneous adverse drug reactions" - namely CADR and ADR?

Author Response

Thank you for your review. We have updated the abstract to reflect the entire manuscript content- not just the introduction. 

Reviewer 3 Report

The authors have compiled a informative review on the cutaneous reactions/side effects of various anti-diabetic agents used in clinic.

A detailed and terse table depicting the immunologically mediated Cutaneous Adverse Drug Reactions (CADR) is well represented and provides an overview of the different types of skin reactions.

Cutaneous reactions to different anti-diabetic medications has been provided including insulin which is  important for clinicians while prescribing medications.

Author Response

Thank you for your review.